# Trophic ecology outweighed intrinsic constraints in shaping skull evolution of carnivorous Permian synapsids

Elías Adán Warshaw [1,3] ✉, Suresh Anmol Singh [1,2] & Michael James Benton [1]

The first truly terrestrial apex predators were carnivorous synapsids, which emerged in the Permian over 260 million years ago and evolved against a backdrop of harsh ecological change. In many ways, these predators mirrored feeding modes and evolutionary trends seen in their much later descendants, the flesh-eating mammals; could apparent resemblances indicate evolutionary constraints on form, or were they shaped by natural selection? Here we show that the skulls of carnivorous Permian synapsids were shaped primarily by adaptation, their shapes reflecting trophic function, and with similarities between distant relatives arising by convergence through natural selection. Conversely, we find no evidence for constraint in terms of the direction or rate of evolution based on patterns of topological modularity. These findings illustrate methods of identifying evolutionary processes in deep time and emphasise the role of competition and adaptation over intrinsic constraints in macroevolution.

The late Carboniferous (Kasimovian, 307–303.7 million years ago (Ma)) saw a drastic restructuring of terrestrial ecosystems[1], with increasing global aridity driving the collapse of the vast coal forests that characterised contemporary equatorial environments, devastating terrestrial vertebrate assemblages which had theretofore maintained a strong reliance on water[2,3]. This crisis set the stage for conquest of the terrestrial realm by amniotes (the least inclusive clade containing reptiles and mammals)[4], with several clades diversifying rapidly to fill ecospace unavailable to taxa more physiologically reliant on available water[2,5]. Principal among these were the synapsids, including the ancestors of mammals, which would go on to dominate terrestrial ecosystems throughout the Permian (299–252 Ma) in both abundance and diversity[6].

The first of two main non-mammalian synapsid radiations occurred from the late Carboniferous to early Permian[7], consisting of pelycosaur-grade taxa including the emblematic sail-backed *Dimetrodon*, a sphenacodont carnivore. This radiation included some of the first amniote herbivores, edaphosaurids and caseids, as well as a plethora of carnivorous lineages[8]. The latter were largely characterised by gracile jaws well suited to rapid apprehension of small prey items[5], carried to an extreme in ophiacodontids and varanopids, but contrasted with a relatively more robust feeding apparatus in some sphenacodontids. Facultative piscivory has been suggested for various basal synapsids[5,9], and even semiaquatic life habits for some (e.g., *Ophiacodon*[8,10]), reflecting persistent reliance on aquatic input among even the most terrestrial synapsid assemblages known from the early Permian[9].

The second radiation, occurring after the extinction of most basal synapsid clades during Olson's Extinction at the end of the early Permian[7,11], was that of Therapsida, the synapsid clade from which mammals are derived. Echoing the earlier radiation of basal synapsids, the ecological diversity of mid-late Permian therapsids extended across multiple trophic levels, including within therapsid subclades; Dinocephalia, for example, evolved both megaherbivorous and large-bodied predatory forms[12]. Carnivorous therapsids were particularly diverse, including repeated convergence on hypertrophied caniniform teeth reminiscent of those of Cenozoic sabre-toothed cats (present in Dinocephalia, Biarmosuchia, Gorgonopsia, and Therocephalia[13,14]). Indeed, beyond morphological similarities, synapsid evolution during this interval records the origin of many hallmarks of extant terrestrial ecosystems, such as communities based around terrestrial primary producers with specialised herbivores and predators (Type II communities sensu Olson[9]). Understanding the mechanisms underlying the assembly of these communities is therefore key to contextualising extant terrestrial ecosystems within a historical (i.e. evolutionary) framework.

A comprehensive view of terrestrial ecosystem evolution through the Permian is precluded by the nature of the fossil record, given that direct evidence of ecological interactions (e.g. predator-prey antagonism) is rarely preserved. However, signatures of large-scale ecological trends are recorded in the functional morphology of their constituent species. This is particularly true for predatory taxa, which play key roles in ecosystem stability[15] and act as agents of selection both directly through predation on prey species and

[1]School of Earth Sciences, University of Bristol, Life Sciences Building, Tyndall Avenue, Bristol, UK. [2]School of Environment, Earth & Ecosystem Sciences, The Open University, Walton Hall, Milton Keynes, UK. [3]Present address: Centre for Integrative Anatomy, Department of Cell and Developmental Biology, Anatomy Building, University College London, London, UK. ✉e-mail: elias.warshaw.25@ucl.ac.uk

indirectly through the establishment of intra-guild competition[16]. Documenting trends in the functional morphology of carnivorous synapsids, the Permian's preeminent predators, may thus unveil details of contemporary ecosystem structure otherwise unresolvable from available fossil data.

This line of study was recently pursued by Singh et al.[5], who analysed ecomorphological evolution in the mandibles of predatory synapsids through the Permian using linear and geometric morphometrics. These authors identified a complexification of terrestrial ecosystems through the Permian, with a major morphofunctional shift occurring between basal synapsids and therapsids. Therapsids were found to inhabit a greater breadth of trophic ecologies relative to basal synapsids, evolving divergent specialisations in either biting power or speed from ancestral ecomorphologies centred around apprehension of prey rather than infliction of damage. Morphofunctional distinctions between basal synapsids and therapsids thus parallel those between sauropsid and mammalian carnivores[5], with apparent niche packing of therapsid-dominated carnivore assemblages further reminiscent of high interspecific competition between ecologically similar carnivores in mammal-dominated ecosystems today[5,17].

A key question emerges from all such macroevolutory analyses: are structures and functions moulded primarily by adaptation or by constraint (i.e. limitations imposed by evolutionary history, material properties, or development)? Repeated convergence upon complex functions between distinct lineages, for example, may be interpreted either as a response to similar selective pressures (highlighting the potency of selection) or as repeated conformity to intrinsic paths of evolutionary least resistance (indicating a limit to the ability of selection to overcome such intrinsic constraints).

The dichotomy between these interpretations mirrors a similar divide among evolutionary biologists, as debates regarding the macroevolutionary importance of adaptation versus intrinsic constraints have permeated the literature for decades. Indeed, this aspect of the relationship between form and function has been a subject of debate since before the time of Darwin[18], but has seen a resurgence of interest in recent decades as palaeontologists have sought to reconcile the fossil record with the principles of population genetics and wider understanding of evolutionary mechanisms[18–21].

Here, we contribute additional evidence to this debate by weighing the relative contributions of structural, historical, and functional constraints on cranial shape evolution in carnivorous Permian synapsids. The cranium is both morphologically complex and functionally diverse[22], playing important roles in protecting the brain, collecting and processing sensory information, and capturing and processing food, to name a few. These characteristics make it an ideal target for studying the evolution of form and function, as cranial shape evolution is subject to a variety of structural as well as functional constraints. By quantifying cranial shape, functional morphology in the context of trophic ecology, and proxies for structural and historical constraints across carnivorous synapsids of the Permian, we examine the assembly of amniote-dominated terrestrial ecosystems through an evolutionary perspective, considering both the patterns present in the fossil record and the evolutionary mechanisms that may have underlain their formation.

## Results & discussion
### Cranial modularity
We used anatomical network analysis (AnNA) to identify topological modules within the skull and mandibles of carnivorous synapsids (i.e. regions of the skull where connections between bones are denser than they are between such regions). These modules arise from the shared evolutionary and developmental histories between bones, as well as the structural demands of the skull[22], making modular boundaries a suitable proxy for inherited (i.e. historical and structural sensu Seilacher[23]) constraints on skull shape evolution.

Anatomical network analysis revealed a relatively consistent pattern of topological modularity across sampled taxa (Fig. 1), with the facial skeleton consistently segregated into anterior and posterior modules. Topological modules in sampled Permian synapsids are highly reminiscent of developmental modularity recorded in placental mammals, among which the anterior portion of the skull is composed of bones embryologically derived from paraxial mesoderm, and the posterior portion is composed of bones derived from neural crest cells[24]. This suggests that the embryological origins of cranial bones in Permian synapsids played a dominant role in influencing the topological organisation of the skull, as opposed to other developmental factors (e.g., timing of ossification, demonstrated to correlate with mode of ossification in mammals[25]).

Notably, we recovered substantial asymmetry between contralateral modules, with midline structures (e.g. the palate or braincase) being assigned to modules of one side of the skull, and paired elements differing in their modular identities on either side of the skull. In *Dimetrodon*, for example, the braincase was assigned to the left temporal module, and the jugals differ in being assigned to the anterior (on the left) or posterior (on the right) modules (Fig. 1). Modular asymmetry has also been observed in anatomical networks of several other amniote groups, where it has been interpreted by different authors as reflecting functional plasticity[26], adaptation for asymmetric utilisation of constituent anatomical parts[27], or an artefact of the dichotomous branching procedure utilised by AnNA[28]. In support of modular asymmetry representing an artefact of AnNA, midline structures not forming their own unpaired modules in our sample were exclusively associated with bones of the left side of the skull (braincase and palate in *Inostrancevia*, braincase in *Dimetrodon*, presphenoid in *Ophiacodon*, and vomer in *Procynosuchus*, *Moschorhinus*, and *Biarmosuchus*). Given that all networks were perfectly symmetrical, this suggests that the module to which midline structures were assigned was ultimately determined by the priority of left-side bones within the alphabetised adjacency matrices used to record anatomical data. We therefore interpret modular asymmetry as reflecting equivocal support for the inclusion of certain elements within multiple modules (left/right for midline structures, anterior/posterior for the jugals in *Dimetrodon*), with midline structures de facto assigned to left-side modules as an artefact of the AnNA procedure.

No phylogenetic trends are apparent in modularity or modular partitions; for example, $Q_{max}$, a measure of modularity, is moderately higher in basal synapsids than in derived therapsids, but these differences do not exceed expected error between most pairs of taxa (Table 1). While more S-modules were recovered for basal synapsids than therapsids (except *Titanophoneus*), all therapsids fit within the range in number of Q-modules exhibited by basal synapsids (six to eight) (these values represent different statistical approaches to identifying topological modules).

Other parameters relating to skull network structure show a similar absence of any clear phylogenetic pattern, albeit with some differences between basal synapsids and therapsids. Variation in the clustering coefficient ($C$; measuring network integration[29]) and connection heterogeneity ($H$) is higher in therapsids than basal synapsids, with values for basal synapsids falling within the range displayed by therapsids (Table 1); the inverse is true for the parcellation index ($P$; measuring the uniformity of modularity across the skull network[30]). The only parameters in which sampled basal synapsids do not overlap with sampled therapsids are $N$ (number of bones) and $L$ (mean shortest path length) (both higher in basal synapsids), reflecting less integrated and thus more loosely connected skull networks in basal synapsids[29]. To this point, relative density of connections ($D$) in all basal synapsid skull networks is lower than in any therapsid except *Titanophoneus* (in which $D$ is still higher than in *Dimetrodon*, which has the lowest density of interosseous contacts in any skull network sampled). Total number of interosseous contacts ($K$) is higher in basal synapsids than in most therapsids (except for *Biarmosuchus*, which overlaps with basal synapsids), which probably reflects higher $N$ in this group given the relative lack of density in these contacts. Our results therefore show a general tendency for basal synapsids to have less integrated skull networks with a greater number of constituent parts than therapsids, while simultaneously

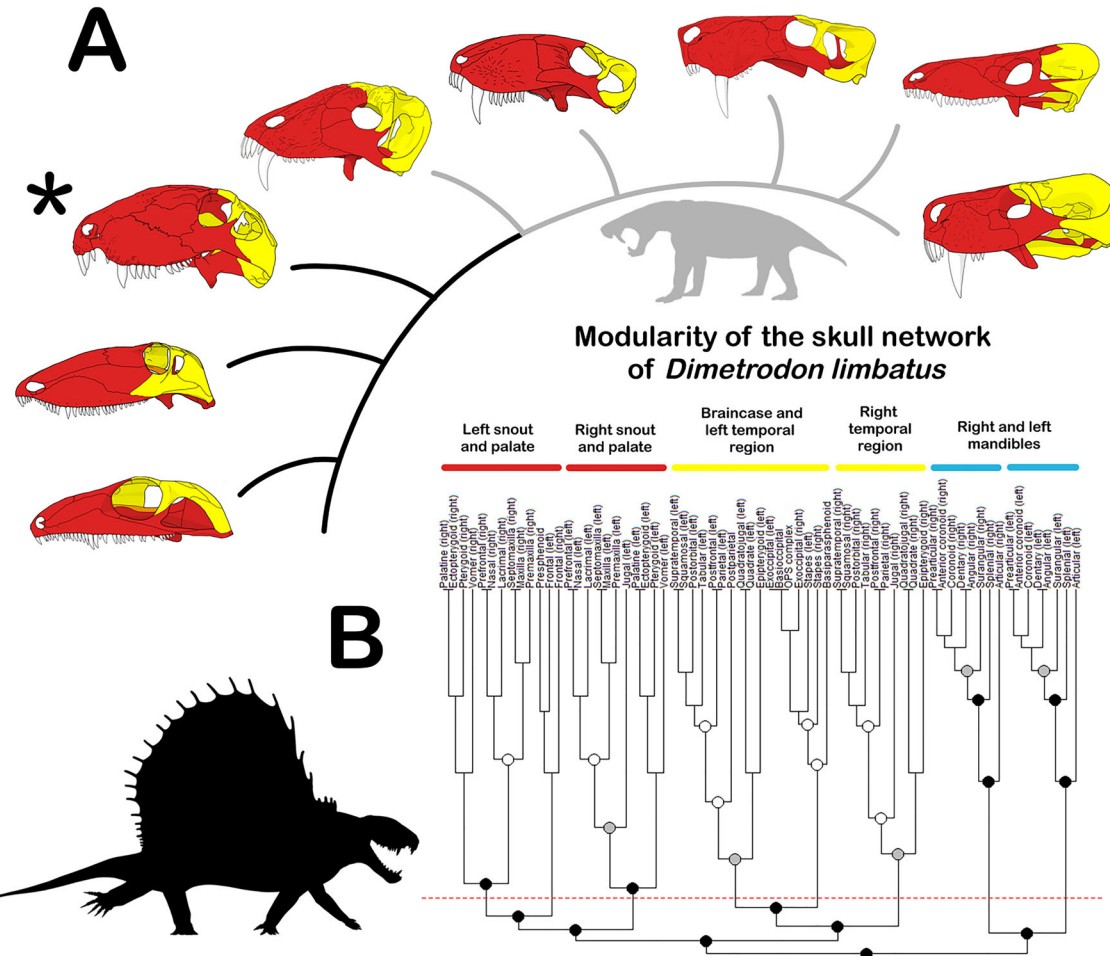

**Fig. 1 | Results of AnNA. A** Skulls and phylogenetic relationships of taxa subjected to AnNA, with S-modules denoted by colour. Clockwise from bottom: *Varanops brevirostris* (redrawn from Romer & Price[8]); *Ophiacodon uniformis* (redrawn from Romer & Price[8]) *Dimetrodon limbatus* (asterisked; redrawn from Romer & Price[8]), *Titanophoneus potens* (redrawn from Orlov[52]), *Biarmosuchus tener* (redrawn from Tatarinov & Rozhdestvensky[54]), *Inostrancevia alexandri* (redrawn from Ivakhnenko[53]), *Moschorhinus kitchingi* (redrawn from Durand[56]), and

*Procynosuchus delaharpeae* (redrawn from Kemp[57]). Skulls are not to scale. Grey branches represent Therapsida; representative silhouette is of *Inostrancevia*.
**B** Example dendrogram showing cranial modularity in *Dimetrodon limbatus*. Node colour denotes the significance level of modular partitions (white: $p < 0.05$; grey: $p < 0.01$; black: $p < 0.001$; $n = 67$ bones); red dotted line denotes Q-module partition. Labelled groups represent S-modules. *Dimetrodon* silhouette from phylopic courtesy of Scott Hartman. OPS Opisthotic/prootic/supraoccipital.

showing marked consistency in patterns of topological organisation (i.e., the positions of modular boundaries) and non-linearity in the evolution of several relevant parameters (see Methods for further explanation of the network parameters produced by AnNA).

## Cranial shape

We quantified shape using a two-dimensional geometric morphometric (2D GM) approach, recording shape using anatomical landmarks placed at homologous points across the crania of sampled specimens in lateral view. Primary axes of cranial shape variation recorded by 2D GM were summarised using principal components analysis (PCA), with a morphospace constructed using the first two principal components (PCs).

The first two PCs describe a cumulative 52% of variance in shape (Fig. 2). PC1 largely records the depth of the skull, while PC2 records the relative sizes of the orbit, rostrum, and temporal region. Therapsids show a much greater distribution in morphospace than basal synapsids, although this is more pronounced on PC2 (where basal synapsids are restricted to higher values, indicating long rostra, large orbits, and small temporal regions).

We identified the first five PCs as significant by identifying a break in the slope of a scree plot (the broken stick method[31,32]), representing a cumulative 80.4% of shape variation (Supplementary Fig. 1).

## Cranial feeding function

Following Singh et al.[5,33], we quantified trophic ecology by scoring sampled specimens for various craniodental characteristics relevant to prey capture and food processing (Supplementary Fig. 2). Missing data was phylogenetically imputed under a kappa evolutionary model, as this was the best-fitting model for functional data from specimens scored for all characteristics (see Methods; Supplementary Table 1). We then used a consensus clustering approach[5,33] to identify three objective functional feeding groups (FFGs) within the dataset; the characteristics defining these groups are provided below, along with their taxonomic constitutions.

**Speed specialists.** Members of this FFG display low mechanical advantage at the anterior end of the toothrow as well as at the canine, with a high posterior mechanical advantage facilitated by a posterior expansion of the toothrow. Additionally, temporal width, premaxillary width, and the angle of the premaxillary alveoli are consistently low among members of this FFG. Only basal synapsids are represented within this FFG, but it does not include all basal synapsids (see below).

**Power specialists.** Power specialists are characterised by increased breadth across the temporal region and premaxillae, as well as pronounced heterodonty including hypertrophied canines and long post-

**Table 1 | Skull network parameters and modularity values generated by AnNA**

| Taxon | N | K | D | C | L | H | P | S-Modules | Q-Modules | $Q_{max}$ | $Q_{max}$ error |
|---|---|---|---|---|---|---|---|---|---|---|---|
| *Varanops brevirostris* | 67 | 163 | 0.074 | 0.471 | 3.773 | 0.392 | 0.813 | 6 | 6 | 0.627 | 0.033 |
| *Ophiacodon uniformis* | 67 | 167 | 0.076 | 0.49 | 3.776 | 0.38 | 0.838 | 7 | 7 | 0.621 | 0.033 |
| *Dimetrodon limbatus* | 67 | 161 | 0.073 | 0.499 | 3.794 | 0.397 | 0.78 | 6 | 8 | 0.635 | 0.033 |
| *Titanophoneus potens* | 62 | 139 | 0.074 | 0.472 | 3.653 | 0.451 | 0.814 | 6 | 6 | 0.588 | 0.038 |
| *Biarmosuchus tener* | 65 | 163 | 0.078 | 0.511 | 3.586 | 0.41 | 0.816 | 5 | 6 | 0.599 | 0.033 |
| *Inostrancevia alexandri* | 61 | 151 | 0.083 | 0.506 | 3.653 | 0.423 | 0.821 | 5 | 7 | 0.563 | 0.037 |
| *Moschorhinus kitchingi* | 62 | 147 | 0.078 | 0.412 | 3.757 | 0.439 | 0.816 | 5 | 7 | 0.543 | 0.035 |
| *Procynosuchus delaharpeae* | 61 | 154 | 0.084 | 0.449 | 3.433 | 0.308 | 0.82 | 5 | 7 | 0.599 | 0.035 |

See text for explanations of calculated values.

incisor diastemata. This is the only FFG to only include therapsids, with representatives of every sampled therapsid subclade. Dinocephalians, biarmosuchians, and gorgonopsians form the majority of this FFG, however.

**Generalists**. Generalist taxa are intermediate in several functional characteristics between speed and power specialists, including skull width, canine length, toothrow length, and angle of the premaxillary alveoli. This FFG is also characterised by high mechanical advantage at the canine and the anterior end of the toothrow. Both therapsids and basal synapsids are represented among generalists, including most sampled therocephalians and cynodonts.

A single taxon, the therocephalian *Gorynychus masyutinae*, could not be allocated to any FFG, as it was recovered in a different group by each clustering algorithm.

As for shape data, we summarised and visualised variation in functional character data using PCA. The first two function PCs describe a total of 55% of sampled variance, with each FFG showing subequal distributions across these axes. No trends are apparent between skull size and function within FFGs, although generalists tend to be smaller than either power or speed specialists (Fig. 2; Supplementary Fig. 3). Shape and function PC results are depicted side by side to emphasise similarities in clade- and FFG-specific patterns between morpho- and function-space. Similarly to shape, therapsids show greater distribution along functional PCs than basal synapsids, although when separated by time this pattern appears to be a result of a wandering mean rather than standing diversity in functional morphology (Fig. 3).

**Temporal patterns in cranial evolution**
Several overarching trends are apparent in the cranial ecomorphology of carnivorous synapsids through the Permian. The most evident of these represent differences between basal synapsids and therapsids, and as such show sharp breaks at Olson's Extinction rather than continuous change. For example, therapsids show markedly broader skulls, shorter tooth rows, and more pronounced heterodonty than basal synapsids on average, echoing the distinctions between the speed specialist and generalist and power specialist FFGs. Otherwise, temporal trends in ecomorphology within therapsids or basal synapsids are difficult to discern from our results (Fig. 2 and 3), and any apparent trends may simply reflect sampling bias (e.g., better separation in function- and morphospace between generalists and power specialists after the end-Capitanian Extinction, which could be an artefact of low sample size at each individual time slice; Fig. 3).

**Phylogenetic signal**
We recovered a moderate but significant phylogenetic signal (λ) within Procrustes-transformed shape coordinates across all sampled synapsids (Table 2). Phylogenetic signal decreased in strength when evaluated within successively less inclusive clades, culminating in a nonsignificant signal within both therapsid subclades with large enough sample sizes to permit

individual tests (Gorgonopsia and Therocephalia), as well as within basal synapsids. This pattern was accentuated when the first five shape PCs were used in place of coordinate data, with a relatively high λ across all taxa contrasting with low (albeit significant) values of λ optimised for therapsids and gorgonopsians, and nonsignificant values for basal synapsids and therocephalians. We found a similar pattern in functional data, but skull size (centroid size from 2D GM) showed high values of λ across all taxonomic levels (not significant within basal synapsids or therocephalians, possibly due to these groups having the lowest sample sizes of any tested).

The increase in phylogenetic signal between shape coordinates and PC scores may reflect an artefact of how PCA interacts with phylogenetic comparative data. Even under a strict multivariate Brownian Motion (BM) model of evolution, some traits (axes of shape variation, in this case) will diverge more than others during the early history of clades for stochastic reasons[34]. Because this divergence is inherited by all succeeding members of the clade, variation along these axes will usually represent the largest proportion of sampled variance and thus be identified by PCA as the primary axes of variation (the first few PC axes). Increased phylogenetic signal in PC scores relative to coordinate data, then, reflects the isolation of traits most characteristic of the distinctions between clades.

The diminution of phylogenetic structure at lower taxonomic levels is a separate pattern, present across both shape coordinates and PC scores as well as functional data, which requires additional explanation. Similar patterns have been identified in other geometric morphometric datasets. For example, Jones and Goswami[35] recovered a strong phylogenetic signal in shape across families of extant pinnipeds (Phocidae and Otariidae) but found that this signal disappeared when analyses were restricted to within-family shape variation. These authors argued that this pattern could be explained by ecological convergence within Phocidae, and by low diversity in cranial shape among otariids. Given that the same pattern is present in both shape and functional morphology of carnivorous Permian synapsids (Table 2), the former explanation seems to better fit the results presented here (although low diversity in skull shape is also a possible cause for low phylogenetic signal in basal synapsids, which inhabited less function- and morphospace than did therapsids; Fig. 2). Ecomorphological evolution within this group, then, was characterised by phylogenetically structured segregation of clades in morpho- and function-space, followed by persistent convergence among taxa within these segregated areas.

This pattern of phylogenetic structure has strong implications for interpreting the selective regimes under which carnivorous Permian synapsids evolved. Simulations have demonstrated that most scenarios of trait evolution have inconsistent effects on phylogenetic signal[36], with variation in evolutionary parameters (e.g. evolutionary rate) causing stark variation in resulting phylogenetic structure. A notable exception is divergent selection, which in simulations performed by Revell et al.[36] was the only evolutionary scenario to consistently result in low phylogenetic signal. Low phylogenetic signal in shape and function among both gorgonopsians and therocephalians therefore suggests that divergent selection was responsible for this pattern, as under most evolutionary scenarios, variation in

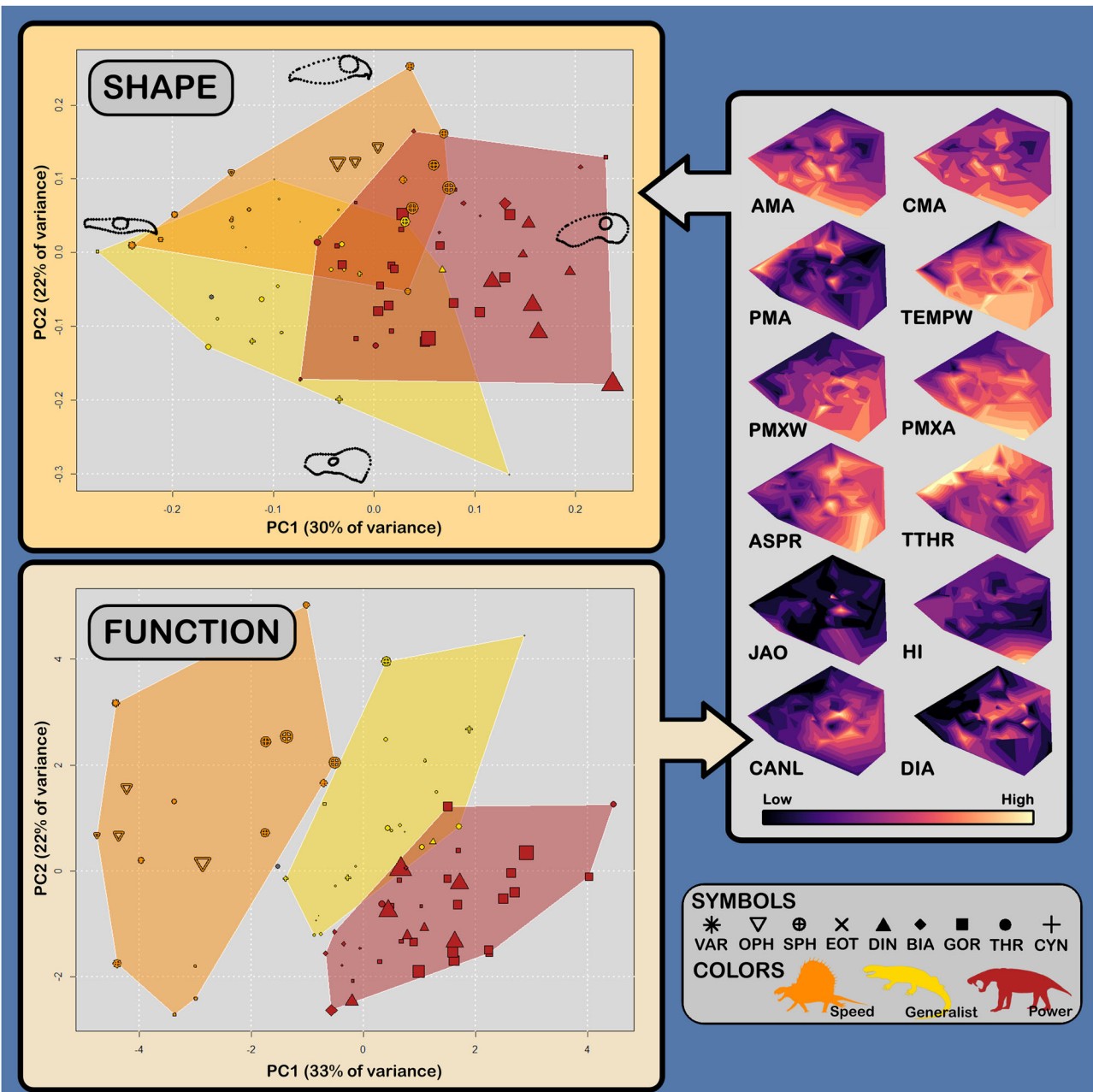

**Fig. 2 | Results of PCA on shape and function.** Skull outlines on shape PC axes show shapes corresponding to minimum and maximum PC values. Points are coloured by FFG designation and are sized proportionally to centroid size (*n* = 77 specimens). Inset at right shows interpolations of functional characteristics onto morphospace (i.e., shape PC plot), with lighter colours denoting higher values for functional measurements; see Methods for explanations of measurements taken. *Dimetrodon* and *Eothyris* silhouettes from phylopic courtesy of Scott Hartman and Nobu Tamura, respectively. AMA anterior mechanical advantage, ASPR maximum aspect ratio, BIA Biarmosuchia, CANL canine length, CMA canine mechanical advantage, CYN Cynodontia, DIA diastema length, DIN Dinocephalia, EOT Eothyrididae, GOR Gorgonopsia, HI homodonty index, JAO jaw articulation offset, OPH Ophiacodontidae, PMA posterior mechanical advantage, PMXA angle of pre-maxillary tooth row, PMXW premaxilla width, SPH Sphenacodontia, TEMPW temporal width, THR Therocephalia, TTHR tooth row length, VAR Varanopidae.

evolutionary parameters between these groups would alter resulting phylogenetic signal.

Low phylogenetic signal can thus be interpreted as resulting from both convergence and divergence, an apparent contradiction necessitating further explanation. Although convergence and divergence are nominally antonymic, the distinction between them is in reference point rather than the evolutionary process they describe. For example, while the therocephalian *Moschorhinus* diverged from other members of this group in evolving hypertrophied canines and a mediolaterally broad skull, the same characteristics made it convergent with many derived gorgonopsians (e.g. rubidgeines[13,37]); divergence relative to sister taxa can thus result in

convergence with more distant relatives. The existence of such a pattern has long been recognised even within therapsid subclades: gorgonopsians, for example, include two convergent examples of robust, large-bodied sabre-toothed taxa (*Inostrancevia* and rubidgeines) which diverged from their respective sister taxa in these regards[37,38].

Ecomorphological convergence between distantly related taxa as a result of divergence at the point of diversification aligns well with expectations set by a hypothesis of high interspecific competition and persistent selective control of ecomorphology. The pattern of phylogenetic structure recovered here within carnivorous synapsid ecomorphology thus enables inference of ecological dynamics within the carnivore guild through the

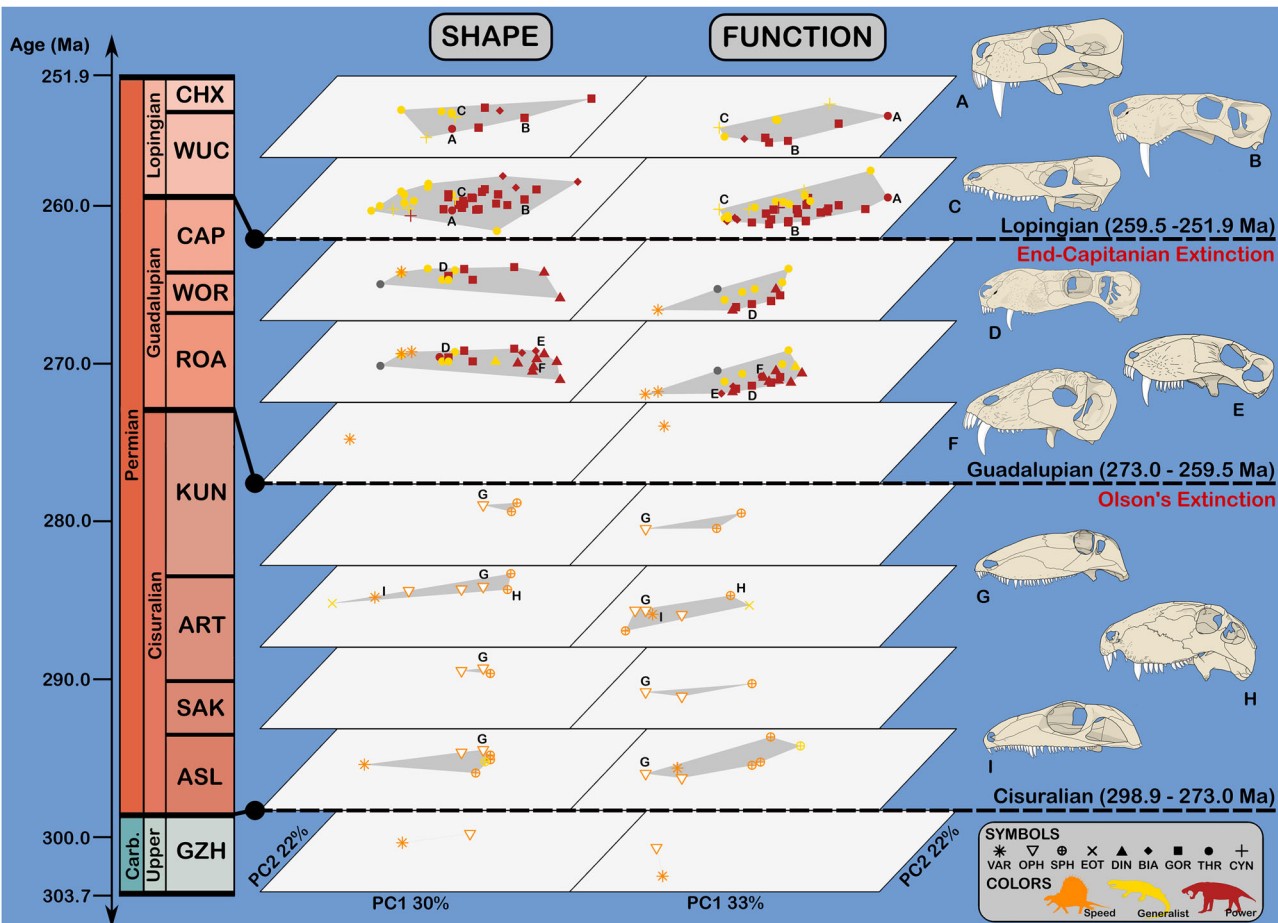

**Fig. 3 | Shape and function in carnivorous synapsids through the Permian.** Each plot is of PC1 and PC2 for shape or function, separated by geological stage (*n* = 77 specimens). Note that taxon ranges are stochastically estimated based on single specimens and therefore not necessarily accurate to the totality of available fossil material (see Methods); additionally, as only tip data is plotted, some stages show artefactually small function- and morphospaces due to sampling biases (e.g. Gzhelian and Roadian). **A** *Moschorhinus kitchingi*; **B** *Inostrancevia alexandri*; **C** *Procynosuchus delaharpeae*; **D** *Suchogorgon golubevi*; **E** *Biarmosuchus tener*; **F** *Titanophoneus potens*; **G** *Ophiacodon uniformis*; **H** *Dimetrodon limbatus*; **I** *Varanops brevirostris*. Skulls are not to scale. Sources for illustrations and silhouettes in legend are as in Fig. 1; abbreviations are as in Fig. 2.

**Table 2 | Phylogenetic signal in shape, function, and skull size**

| Clade | n | Shape (coordinates) | | Shape (PCs 1–5) | | Function | | Size | |
|---|---|---|---|---|---|---|---|---|---|
| | | λ | p | λ | p | λ | p | λ | p |
| Synapsida | 77 | **0.372** | **0.001** | **0.882** | **0.001** | **0.621** | **0.001** | **0.977** | **0.005** |
| Basal Synapsida | 19 | 0.325 | 0.753 | 0.600 | 0.079 | 0.166 | 0.246 | 1.000 | 0.731 |
| Therapsida | 58 | **0.058** | **0.001** | **0.309** | **0.001** | **0.169** | **0.001** | **0.665** | **0.001** |
| Gorgonopsia | 21 | 0.115 | 0.901 | **0.302** | **0.003** | 0.059 | 0.129 | **0.859** | **0.011** |
| Therocephalia | 18 | 0.213 | 0.250 | 0.153 | 0.817 | 0.244 | 0.390 | 0.802 | 0.137 |

Significant signals are bolded (α = 0.05). Values for Gorgonopsia and Therocephalia are shown to the exclusion of other therapsid subclades because they have the highest sample sizes among sampled synapsid clades.

Permian. It is notable that most examples of convergence in form and function occur between taxa separated in either time or space; for example, *Inostrancevia* and its closest relatives first appear in the fossil record in Laurasia whereas rubidgeines are known only from southern Gondwana[38]. Convergence in these cases would therefore not have resulted in direct competition (although *Inostrancevia* later coexisted with rubidgeines in the Ruhuhu Basin of Tanzania[39], long after their respective origins). This suggests that divergence in ecomorphology was driven by selective pressures imposed by interspecific competition, such that diverging taxa minimised

competition with each other and maximised their ability to exploit available resources.

It should be emphasised that FFGs are not likely to correspond directly to prey choice without consideration of other factors (i.e. all members of a single FFG were unlikely to have preferred the same prey). Rather, FFGs reflect biomechanical optimisations for particular methods of food procurement (including prey capture, killing mode, and carcass processing). Optimal prey choice according to a given method must also have been controlled by body size. Therefore, morphologically similar and coeval taxa

**Table 3 | Correlations between shape, function, and skull size**

| Clade | n | Shape~function | | Shape PCs~function | | Shape~size | | Function~size | |
|---|---|---|---|---|---|---|---|---|---|
| | | R² | p | R² | p | R² | p | R² | p |
| Synapsida | 77 | **0.335** | **0.001** | **0.428** | **0.001** | **0.066** | **0.001** | 0.039 | 0.065 |
| Basal Synapsida | 19 | 0.688 | 0.423 | **0.900** | **0.001** | **0.332** | **0.001** | 0.098 | 0.146 |
| Therapsida | 58 | **0.341** | **0.001** | **0.493** | **0.001** | **0.048** | **0.006** | 0.010 | 0.493 |
| Gorgonopsia | 21 | **0.694** | **0.041** | **0.783** | **0.028** | 0.049 | 0.324 | **0.127** | **0.022** |
| Therocephalia | 18 | **0.846** | **0.013** | **0.870** | **0.030** | 0.067 | 0.303 | 0.059 | 0.407 |

Significant correlations are bolded (α = 0.05). Shape refers to Procrustes-transformed shape coordinates, shape PCs refers to PC scores for shape PCs 1–5, function refers to calculated functional characteristics, and size refers to log-transformed centroid size. Correlations were calculated using linear model fitting, except in the case of phylogenetically structured variables, which were correlated via phylogenetic generalised least squares regression (all correlations for therapsids and all synapsids, and correlations involving shape PCs and skull size for Gorgonopsia). Values for Gorgonopsia and Therocephalia are shown to the exclusion of other therapsid subclades because they have the highest sample sizes among sampled synapsid clades.

of different sizes were unlikely to have preferred similar prey (the badger-sized *Moschorhinus*, for example, is unlikely to have hunted prey as large as the bear-sized gorgonopsians with which it shared its environment despite similar biomechanical specialisations). As noted by Singh et al.[5], robust synapsids of the late Permian (power specialists of this study) show various specialisations concordant with hunting relatively large prey, whereas most basal synapsids and relatively gracile therapsids (speed specialists and generalists) likely preferred prey of smaller size. Power specialists also tended to be larger than generalists (Fig. 2), such that body size emphasised differences in prey choice implied by FFG specialisations and provided an additional avenue for niche partitioning and the minimisation of direct competition among coeval carnivorous synapsids. Counterintuitively, the consistently high (albeit not always significant) phylogenetic signal in skull size breaks from the pattern exhibited by both shape and functional morphology (Table 2), so body size was likely not under the same selective regime as these variables (perhaps reflecting the greater variety of biological functions impacted by body size, e.g. thermal physiology).

**Comparison of shape, function, and modularity data**
Our analyses highlight a perceptible influence of internal constraints on cranial shape evolution in carnivorous Permian synapsids. Allometry, for example, explains a significant amount of variance in skull shape among all sampled taxa, basal synapsids, and therapsids (Table 3; the amount of shape variance explainable by allometry is quite low in therapsids and across all synapsids, but substantial within basal synapsids). The influence of topological modularity (or more likely, developmental modularity as reflected by topological modules) is more nuanced. Topological modules show significant covariation in shape ($r_{PLS} = 0.939$; $p = 0.001$), indicating integration of skull shape evolution across modular boundaries. However, within-module shape covariation is also significant (CR = 0.836; $p = 0.001$), suggesting that cranial shape evolution in Permian synapsids utilised module-specific developmental mechanisms (producing the modular signal) as well as mechanisms affecting both modules jointly (although not necessarily the entirety of the skull; between-module shape covariation could also be produced by integration of discontinuous regions inhabiting different modules, for example).

Nevertheless, the combined influences of these internal constraints leave much to be explained regarding the rate and pattern of cranial shape evolution. Most conspicuously, relative consistency in topological modularity contrasts heavily with marked differences in function- and morphospace occupation between basal synapsids and therapsids. To this point, the hierarchical nature of phylogenetic structure recovered within shape (suggesting frequent divergence within the bounds of clade-specific regions of morphospace; see above) is not matched by any corresponding phylogenetic hierarchy in patterns of topological modularity (Fig. 1; Table 1). Moreover, within-module shape covariance does not differ significantly between basal synapsids and therapsids ($z = 0.817$; $p = 0.414$), such that hallmarks of therapsid skull morphology (e.g. an expanded temporal region) find no readily apparent explanation as

structural consequences (i.e. spandrels[40]) of a shift in underlying developmental mechanisms (although if they did, the impetus for this shift would similarly demand explanation).

These difficulties in explanation find immediate resolution when adaptation for trophic function is viewed as the primary driver of cranial shape evolution in carnivorous Permian synapsids. We recovered a strong relationship between Procrustes-transformed shape coordinates and functional morphology which mirrors the hierarchical pattern recovered in phylogenetic signal (Table 3), with the strength of the observed correlation increasing at successively lower taxonomic ranks (such that this correlation is tightest within therapsid subclades). While we found no significant correlation between these variables within basal synapsids, R² values increased across all tested clades when shape PCs 1–5 were substituted for shape coordinates, resulting in a strong and highly significant correlation between shape and function for this group (in addition to significant correlations for every other tested taxonomic grouping). This increase suggests that shape change early in the history of tested clades, preferentially sorted into significant PCs (see above), accounts for a disproportionate amount of the functional signal in shape (in addition to the phylogenetic signal), and thus that much of the ecomorphological distance between groups (and to a lesser degree, within them; Table 3) evolved early on. Furthermore, the greater correlation between shape and functional data within therapsid subclades compared to across all synapsids or all therapsids indicates that these groups retained strong but clade-specific relationships between form and function; conversely, substantially higher R² for this relationship in basal synapsids than therapsids probably reflects lower functional diversity in the former group (as also shown by more restricted function- and morphospace occupation in basal synapsids relative to therapsids; Fig. 2).

These results are consistent with those reported for phylogenetic signal, which together paint a picture of cranial shape evolution in carnivorous Permian synapsids driven primarily by adaptation of form to trophic function. This is best illustrated by comparison of our results with the predictions set by alternative scenarios. For example, if developmental constraints represented a major limiting factor on morphospace occupation, then shifts in patterns of topological modularity (our proxy for developmental constraint) would be expected as a prerequisite to the evolution of novel skull shapes (as in therapsids, which show a broader and noticeably distinct distribution in morphospace compared to basal synapsids; Fig. 2). Additionally, if cranial shape and functional morphology both evolved under divergent selection for different reasons (e.g., divergence in trophic ecology for functional morphology, and divergence in visual display structures for cranial shape), then low phylogenetic signal in both of these variables at lower taxonomic levels would be expected in addition to an inconsistent relationship between shape and function data (and thus low R² values for this relationship in all tested groups). Neither of these scenarios find their predictions met in our results, necessitating an alternative explanation of observed patterns in cranial shape evolution being causally related to trophic adaptation.

**Table 4 | Fit of different evolutionary models to cranial shape, function, and size data**

|  | Brownian motion | | Early burst | | Kappa | | Delta | |
|---|---|---|---|---|---|---|---|---|
|  | AIC | lnL | AIC | lnL | AIC | lnL | AIC | lnL |
| Shape | −957.737 | 480.868 | −956.340 | 481.195 | **−1049.629** | **527.815** | −955.820 | 480.910 |
| Function | −3173.396 | 1588.898 | −3173.68 | 1589.840 | **−3403.898** | **1704.949** | −3171.892 | 1588.946 |
| Size | 1194.054 | −595.027 | 1195.930 | −594.965 | **1135.347** | **−564.674** | 1195.796 | −594.898 |

Best performing model is bolded.
AIC Akaike Information Criterion, lnL log-likelihood.

## Evolutionary modelling and rates

Shape, function, and size data all fit best to a kappa model of evolution (Table 4), indicating that the majority of ecomorphological evolution in sampled synapsids occurred near nodes rather than tips (i.e. near points of diversification rather than later within lineages). Prima facie, this would seem to provide evidence for evolution by punctuated equilibria (sensu Eldredge and Gould[21]), as a key tenet of this evolutionary model is a concentration of morphological change at the point of speciation. However, an alternative explanation follows directly from the interpretation presented above of the selective regimes under which carnivorous Permian synapsid ecomorphology evolved: divergent selection at the point of speciation would tend to concentrate evolutionary change at points of diversification, so long as resulting species were subsequently subjected to stabilising selection (e.g. after reaching an adaptive peak)[41]. This explanation predicts a pattern similar to that outlined by the theory of punctuated equilibria (i.e. relatively higher rates of evolution at the point of speciation) but differs markedly from this evolutionary model by regarding apparent stasis as being maintained by selection rather than reflecting constraints on evolution imposed by population structure[21]. Thus, by explaining the concentration of morphological change at the point of speciation as a consequence of divergent selective pressures between recently formed species, we depict cladogenesis as a potent but not necessarily exclusive avenue towards sustained evolutionary changes (given that selective pressures could also change for reasons other than speciation).

The hypothesis that fluctuations in evolutionary rate reflect shifting selective pressures predicts that major faunal reorganisations should be accompanied by elevated evolutionary rates as taxa adapt to fill new niches. There is evidence for such a pattern in the distribution of evolutionary rates in carnivorous synapsids across the Permian (Fig. 4). Guadalupian therapsids show significantly higher rates of skull shape evolution than basal synapsids (observed ratio: 2.534; $p = 0.030$), presumably reflecting their diversification in the wake of Olson's extinction. Lopingian therapsids, sampled well after the establishment of therapsid morphotypes, show intermediate evolutionary rates which cannot be distinguished either from those of basal synapsids (observed ratio: 1.480; $p = 0.237$) or Guadalupian therapsids (observed ratio: 0.584; $p = 0.150$). It is therefore uncertain whether the rate of skull shape evolution in carnivorous synapsids remained higher in therapsids throughout the Permian than in basal synapsids, or if the pace of therapsid skull shape evolution fell to rates comparable to those seen in basal synapsids during the Lopingian. Arguments from theoretical expectations could be made towards either alternative: greater ecological complexity in therapsid-dominated ecosystems, recovered by Singh et al.[5] and reaffirmed by our finding of an increase in FFG diversity after Olson's extinction, could have promoted greater evolutionary rates in therapsids through the creation of a more volatile adaptive landscape. Conversely, evolutionary rates in therapsids would be expected to fall following the stabilisation of ecosystem structure and clade-specific morphospace occupation. Ultimately, more data is required to resolve this point.

## Parallels between carnivorous Permian synapsids and mammalian carnivores revisited

While our results reiterate several parallels between carnivorous Permian synapsids and mammalian carnivores previously recovered by other authors (e.g. high potential for interspecific competition amongst coeval carnivores[5]), they also highlight additional similarities, albeit with some nuance. For example, topological modules recovered here within the skulls of carnivorous Permian synapsids are highly reminiscent of developmental modules in placental mammals, with the anterior portion of the placental skull being composed of bones embryologically derived from paraxial mesoderm and the posterior portion composed of bones derived from neural crest cells[24]. Notably, Goswami et al.[42] found that cranial bones derived from neural crest cells exhibited higher rates of shape evolution in placental mammals relative to those derived from paraxial mesoderm. By contrast, we found no significant difference in the rate of shape evolution between topological modules in carnivorous Permian synapsids (observed rate ratio = 1.117; $p = 0.688$).

This difference in macroevolutionary pattern is unlikely to represent a shift in the capacity of neural crest cell-derived cranial bones for higher rates of evolution between Permian therapsids and placental mammals, given that the multipotency of neural crest cells (interpreted by Goswami et al.[42] as facilitating the rate difference observed in their dataset) is a characteristic common to all vertebrates[43]. Instead, we interpret the absence of any significant difference in the rate of shape evolution recovered between topological modules in carnivorous Permian synapsids to reflect differences in the nature of the datasets and methodology between our study and Goswami et al.[42], as well as a potential ecological signal. First, the topological modules recovered here do not correspond directly to the embryological origins of bones in the placental skull; for example, the squamosal is neural crest cell-derived but included within the posterior cranial module along with the mesoderm-derived parietal[24,42]. Placental mammals are also more densely sampled than are Permian therapsids (compare for example the 322 specimens analysed by Goswami et al.[42] to the 77 specimens included in our analyses), largely owing to their representation within the modern biota. Coarse sampling could result in genuine rate differences between modules being obscured, especially given that these differences in placental mammals are belied by roughly equivalent amounts of morphological variation in either mesoderm or neural crest cell-derived bones[42]. However, it should be noted that many escalations in the rate of placental skull shape evolution were associated with the acquisition of new ecological specialisations which find no counterpart in Permian therapsids (e.g. transitions to aquatic lifestyles in cetaceans and pinnipeds[42]). Apparent equivalence in evolutionary rate between Permian synapsid cranial modules may therefore reflect the smaller range of ecospace explored by this group relative to placental mammals, as rates of shape evolution calculated from fossils record long-term averages of much faster instantaneous rates[44,45], which will appear slower if species are stabilised around established morphotypes rather than being directed into new regions of morphospace.

Another parallel between mammals and Permian synapsids highlighted by our results is the macroevolutionary patterns which have been recorded in either group. Various authors have identified a significant increase in ecological disparity among placental mammals following the end-Cretaceous mass extinction[46–48], signifying an adaptive radiation into ecospace made available by the extinction of the non-avian dinosaurs

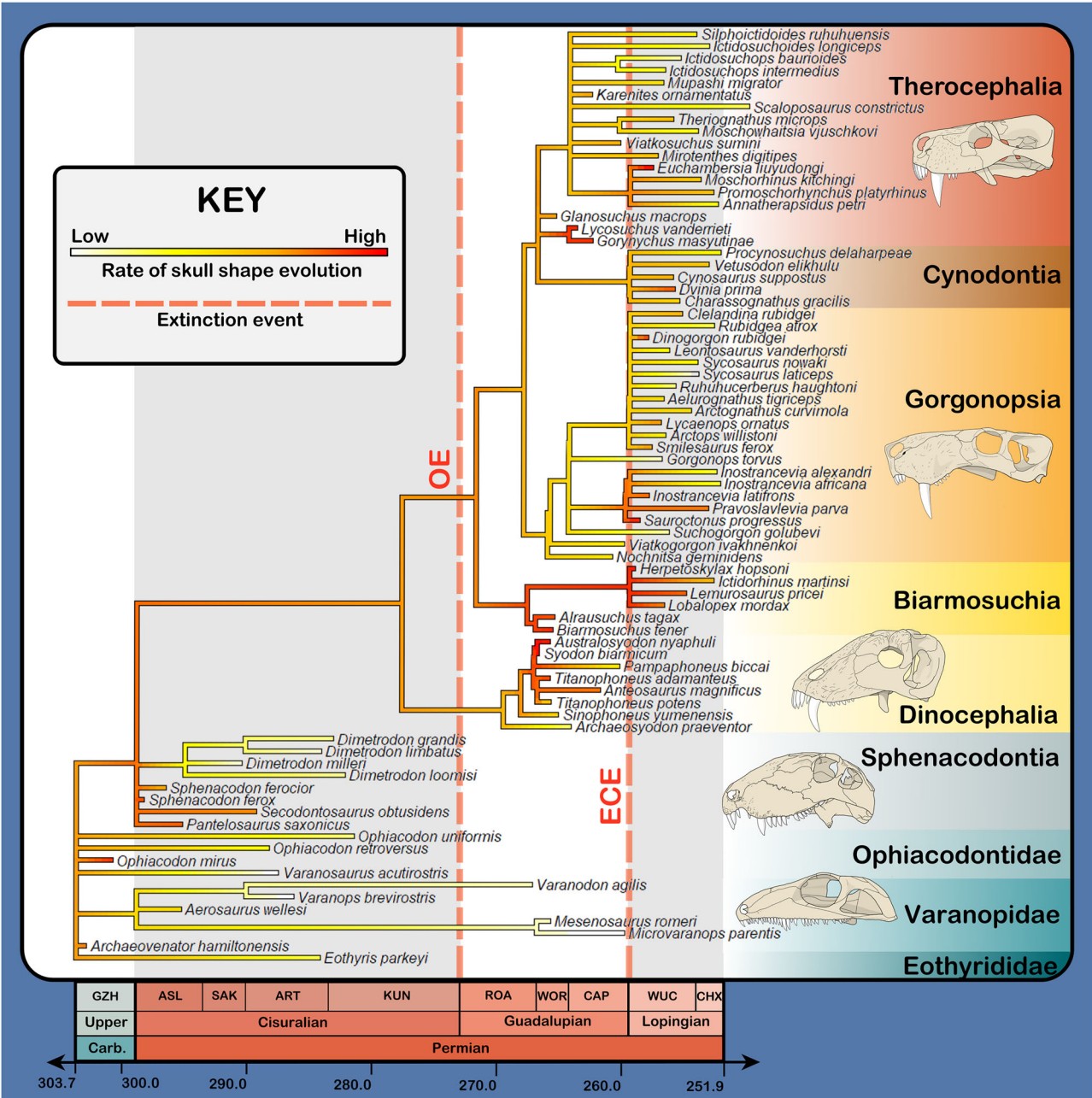

**Fig. 4 | Rates of skull shape evolution across carnivorous Permian synapsids.** Tree includes all analysed taxa (*n* = 77 species). Skulls at right are, from top to bottom: *Moschorhinus kitchingi, Inostrancevia alexandri, Titanophoneus potens,* *Dimetrodon limbatus, Varanops brevirostris*. Skulls are not to scale. Sources for illustrations are as in Fig. 1. OE Olson's extinction, ECE End-Capitanian extinction.

(among other groups). Similar adaptive radiations in mammals have been recognised in tandem with major ecological disturbances throughout the Cenozoic, including the spread of grasslands in the Miocene[49]. Indeed, Simpson[19] highlighted the rapid establishment of adaptive zones in felids and equids around this interval (followed by a restriction of ecomorphological evolution to within these zones) as archetypal examples of quantum evolution. This pattern of rapid change in response to environmental perturbation followed by ecological stability has additionally been recognised within various fossil groups as serial examples of Court Jester and Red Queen evolution, respectively[50], further highlighting the ubiquity of the patterns recovered here within carnivorous Permian synapsids.

These similarities in evolutionary mode are unlikely to follow from intrinsic constraints common to Permian synapsids and mammals, given

that we found little evidence for intrinsic constraints on the pattern of skull shape evolution in carnivorous Permian synapsids. They are therefore better understood as the signature of common extrinsic evolutionary forces: namely, adaptation by natural selection. A close association between trophic ecology and skull shape at the macroevolutionary scale has also been noted in placental mammals and explained by the complexity of mammalian feeding[42], which may similarly explain the intimate relationship between feeding function and skull shape as synapsids evolved precursors to the mammalian feeding apparatus throughout the Permian. Notably, this explanation centres the ecological relevance of traits as a predictor for their macroevolutionary relevance. The importance of selection on traits at the organismal level in shaping broad patterns in the evolution of both carnivorous Permian synapsids and mammals blurs the distinction between

https://doi.org/10.1038/s42003-026-09824-3                                                                                                                   **Article**

micro- and macroevolution and contrasts with evolutionary models which contend that these different scales of evolutionary change are governed by fundamentally different forces. Furthermore, it supports an understanding of macroevolution as being guided primarily by interactions between organisms and their environments. Parallels in the evolution of groups separated across time and space do not merely represent inherited limitations, but common responses to selective pressures that echo through deep time.

## Methods
### Materials
Photographs or reconstructions of skulls in lateral view were collected from the literature for 77 taxa of synapsid carnivores, including representatives of Therocephalia ($n = 18$), Cynodontia ($n = 5$), Gorgonopsia ($n = 21$), Dinocephalia ($n = 8$), Biarmosuchia ($n = 6$), Sphenacodontia ($n = 8$), Ophiacodontidae ($n = 4$), Varanopidae ($n = 6$), and a single eothyridid (*Eothyris parkeyi*). Specimens were selected on the basis of preservation quality and ontogenetic stage, with juvenile individuals excluded from the sample; only taxa with published phylogenetic placements were sampled (although in the absence of published phylogenetic relationships, genera with multiple species were treated as polytomies). Images in ventral and/or dorsal views were also collected for as many specimens as possible for the collection of linear morphometric data (see below). Phylogenetic relationships and stratigraphic ranges of sampled taxa were sourced largely from Singh et al.[5], with data for 11 additional taxa collected elsewhere from the literature.

### Anatomical network analysis
Anatomical network analysis is a recently developed methodology that integrates graph theory, a mathematical discipline, with the study of anatomical complexity, exploring patterns of connectivity within anatomical systems (e.g., the skull) by representing them as networks of nodes and connections (generally bones and osseous contacts, respectively[22,51]). Networks are then analysed to identify regions where connections between nodes are denser than they are between such regions (topological modules).

In order to chart the evolution of topological modularity among Permian synapsids, we performed AnNA on the skull and mandible of a member of each major sampled clade (except Eothyrididae, for which published anatomical information was insufficient): *Varanops brevirostris* (Varanopidae)[8], *Ophiacodon uniformis* (Ophiacodontidae)[8], *Dimetrodon limbatus* (Sphenacodontia)[8], *Titanophoneus potens* (Dinocephalia)[52,53], *Biarmosuchus tener* (Biarmosuchia)[53–55], *Inostrancevia alexandri* (Gorgonopsia)[53–55], *Moschorhinus kitchingi* (Therocephalia)[56], and *Procynosuchus delaharpeae* (Cynodontia)[57]. Observations were made from published photographs and specimen drawings, as well as textual descriptions, and personal observation of specimens wherever possible (*Dimetrodon* UMZC T.320; *Moschorhinus* NHMUK R5698; *Procynosuchus* NHMUK R37054). Analytical procedure followed recently published applications of AnNA[22,26,51], to which readers are directed for in-depth explanations of the theoretical and mathematical underpinnings of the methodology. We summarise the AnNA procedure below.

Contacts between bones were recorded as adjacency matrices in Excel, with "1" denoting a contact. Matrices were then read into R and used to generate dendrograms representing topological similarity between bones, where proximity between bones in the dendrogram reflects greater numbers of shared neighbours. Dendrograms were partitioned into modules using two separate approaches, resulting in S-modules and Q-modules, respectively. S-modules represent statistically significant ($\alpha = 0.05$) modular partitions as determined by a Mann–Whitney U test, with a null hypothesis of equivalence in connection density within and between modules (i.e., no modular signal). Q-modules, by contrast, represent the modular partition which maximises the anatomical network's Q value, a measure of modularity introduced by Clauset et al.[58] and Newman and Girvan[59], which quantifies modular signal (i.e., density of within-module connections) relative to a randomly distributed network. Expected error for maximum Q

values ($Q_{max}$) was calculated by jack-knifing, treating each connection as an independent observation.

AnNA also produces several parameters that identify key aspects of network structure. The most basic of these are the total numbers of nodes (in this case, bones; $N$) and connections between them (contacts; $K$). Additionally, connection density ($D$) is the ratio between the observed number of connections and the maximum number possible for a given network, thus reflecting relative integration of the network. Similarly, the clustering coefficient ($C$) measures the number of connections among a node's neighbours relative to the maximum possible number of such connections. Mean shortest path length ($L$) records the average minimum distance between all possible pairs of nodes, and together with $D$ and $C$ reflects the complexity and interconnectedness of the network (with highly complex, interconnected networks sharing low $L$ and high $D$ and $C$[61]). Two final parameters, connection heterogeneity ($H$) and parcellation index ($P$), are concerned with the distribution of connections: $H$ measures inter-node variance in number of connections, and $P$ measures the uniformity and degree of modular signal across the network[30].

AnNA was performed in R using the package igraph[60]. R script was sourced from Strong et al.[51], itself modified from Werneburg et al.[26] and Plateau and Foth[28].

### Two-dimensional geometric morphometrics
We scored all sampled skulls for ten landmarks, with a further 113 sliding semilandmarks placed along nine semilandmark curves between landmarks (Supplementary Fig. 2). Landmarks were selected with the goal of maximising the amount of cranial shape recorded while also permitting the inclusion of a broad sample (i.e. without excluding too many taxa because of missing data). To this end, landmarks were placed along the outline of the entire skull (excluding palatal and occipital elements when visible laterally) as well as around the orbital margin. We used two-dimensional rather than three-dimensional data given the greater availability of relevant data in the literature, allowing for a more taxonomically inclusive dataset. Digitisation was undertaken in tpsDig[61], with semilandmarks appended to curves in tpsUtil[62].

Following Procrustes transformation of landmark coordinates to remove non-shape variation (e.g., from size or orientation), we estimated the positions of missing landmarks in R using the function *estimate.missing* from the package geomorph[63]. We then performed a second Procrustes transformation with semilandmarks sliding along curves, and summarised shape variation by subjecting Procrustes-transformed coordinates to Principal Components Analysis (PCA). The dimensionality of shape data was reduced to facilitate evolutionary model fitting (see below) by identifying significant principal components (PCs) through the broken stick method[31,32], wherein a bar plot of cumulative shape variance represented by each PC was examined for a sharp break in slope (Supplementary Fig. 1). Both Procrustes transformations, PCA, and all subsequent analyses of 2D GM data were carried out in R using the package geomorph[63].

We sorted landmarks into modules based on the results of AnNA, with boundaries between the anterior and posterior modules lying at the jugal-squamosal contact ventrally and the posterior margin of the prefrontal dorsally (representing the most common locations for this boundary across analysed taxa – *Ophiacodon* and *Procynosuchus* break from this pattern, albeit slightly; Fig. 1). Within- and between-module covariation in shape was then calculated with correction for phylogeny using the R functions *phylo.modularity* and *phylo.integration*, respectively. Difference in evolutionary rate between modules was calculated and evaluated for statistical significance with the function *compare.multi.evol.rates*. Both functions are from the R package geomorph[64] and were run with 999 iterations for significance testing.

### Functional morphometric analyses
We quantified functional morphology for each sampled specimen using linear morphometrics of 12 characteristics with well-established

relationships to trophic ecology (Supplementary Fig. 2). Reiteration of shape data was minimised by focusing on characteristics excluded from 2D GM, including dental characters and characters not visible in lateral view. Analysed characteristics were as follows:

**Anterior mechanical advantage (AMA).** Mechanical advantage (MA) records the ratio in length of the inlever (moment arm) to the outlever (load arm), reflecting the efficiency of force output relative to input[33]. In the case of jaw biomechanics, the inlever is the distance between the jaw joint and the line of action of the adductor musculature, while the outlever is the distance between the jaw joint and the point at which force is applied along the jaw. The outlever for AMA was measured between the jaw joint and the anterior-most point along the toothrow, providing the longest possible outlever and thus the lowest possible MA. All calculated MAs had outlevers terminating along the toothrow rather than at the tips of teeth, so as to allow measurement of specimens lacking preserved teeth.

**Canine mechanical advantage (CMA).** As in AMA, but with the outlever terminating at the midpoint of the canine alveolus. This provides a measure of biting efficiency at the canine, which plays an important role in dispatching prey among extant synapsid (i.e., mammalian) carnivores and surely did as well in their Permian counterparts[13]. For specimens with two erupted canines (indicating death during replacement of the canine), the outlever was measured from the posterior canine so as to maximise calculated MA.

**Posterior mechanical advantage (PMA).** PMA provides the maximum possible value of MA, with the outlever terminating at the posterior-most point along the toothrow.

**Temporal width (TEMPW).** The temporal region houses the adductor chamber, such that the width of this region serves as a proxy for the size of the adductor musculature and thus the strength of the bite[64,65]. This measurement was size standardised by dividing by total skull length.

**Premaxilla width (PMXW).** The width across the premaxillae serves as a proxy for the resistance of this region to mediolateral stresses associated with grasping struggling prey. This measurement was size standardised by dividing by total skull length.

**Angle of premaxillary tooth row (PMXA).** The incisors play an important role in food processing among extant mammalian carnivores, acting to crop flesh from carcasses[13]. Greater precision in this function is provided by an increased angle of the premaxillary alveoli relative to the sagittal plane, which places the incisors along a different plane than the rest of the toothrow, allowing for them to be engaged separately.

**Maximum aspect ratio (ASPR).** Maximum aspect ratio refers to the greatest depth of the skull, measured here above the tooth row in order to serve as a proxy for flexural resistance to dorsoventral stresses incurred during biting and/or grasping of live prey. This measurement was size standardised by dividing by total skull length.

**Toothrow length (TTHR).** The length of the toothrow determines the range of MA available along different points of the jaw and is related to the relative importance of the dentition in jaw functionality[33]. This measurement was size standardised by dividing by total skull length.

**Jaw articulation offset (JAO).** The offset of the jaw articulation from the tooth row impacts occlusion, with smaller offsets being typical of carnivorous taxa and producing scissor-like occlusion where the posterior-most teeth come into contact first[33]. This measurement was size standardised by dividing by total skull length.

**Homodonty index (HI).** Heterodonty is a hallmark of mammals that began to develop among their Permian ancestors, allowing greater specialisation of tooth function along the toothrow and presumably facilitating concomitant ecological specialisation. Quantifying its inverse, homodonty, by dividing the length of the smallest tooth by that of the largest, allows for the progressive specialisation of the synapsid dentition to be charted.

**Canine length (CANL).** As mentioned above, the canine likely served an important role in dispatching prey among Permian synapsid carnivores, evidenced especially by the repeated evolution of hypertrophied, sabre-like canines[13]. The length of the canine determines both its relative resistance to breakage and its utility for different styles of hunting, making it a reasonable inclusion among the functional characteristics measured here. This measurement was size standardised by dividing by total skull length.

**Post-incisor diastema length (DIA).** Diastemata facilitate functional separation of different regions of the dentition by allowing them to be engaged separately from one another. Diastema length, then, is another proxy for complexity in tooth function. This measurement was size standardised by dividing by total skull length.

We fitted several evolutionary models to a subset of the functional data including only taxa scored for all characters to identify the best evolutionary model for phylogenetic imputation of missing data. Brownian motion, early burst, kappa, and delta models were tested using the function *transformPhylo.ML* from the R package motmot[66]. Phylogenetic imputation of missing data was performed using the R package phylopars[67].

Following Singh et al.[33], we subjected the complete functional dataset to three different clustering algorithms (K-means, partitioning around medioids, and hierarchical clustering) using the R package factoextra[68] in order to identify objective groupings of taxa based on shared functional characteristics (i.e. FFGs).

**Time-scaled phylogeny**

Phylogenetic context and stratigraphic occurrences for sampled specimens were largely taken from Singh et al.[5], with data for additional taxa sourced from elsewhere in the literature. We then created a time-scaled tree using the function *cal3TimePaleoPhy* in the R package paleotree[69], with sampling, diversification, and extinction rates estimated using the function *make_durationFreqCont* and taxon observations treated as random points within input stratigraphic ranges. The accuracy of this tree to current understanding of Permian synapsid clade divergence times was maximised by using dates from a recent rigorously estimated time-calibrated tree[70] to constrain the ages of as many nodes as possible. All polytomies were randomly resolved during the estimation procedure (a requirement for evolutionary model fitting, maintained across all other analyses for the sake of consistency).

**Statistics and reproducibility**

Using the time-scaled tree, we quantified phylogenetic signal in shape, function, and skull size (=centroid size determined by 2D GM) to identify datasets that required phylogenetic correction during comparison. Phylogenetic signal ($\lambda$) was quantified using the function *physignal.z*, with $\lambda$ calculated by taking the mean value optimised across all data dimensions. The relationships between cranial shape, functional morphology and size were then compared using linear model fitting (or phylogenetic generalised least squares regression for independent variables with significant phylogenetic signals). Quantifications of phylogenetic signal and correlations between variables both used 999 iterations for significance testing. Each of these analyses was carried out using the R package geomorph[63].

Species-specific rates of skull shape evolution were isolated by treating each species as a separate clade within the geomorph[63] function *compare.evol.rates*. We also used this function to compare rates of shape evolution between basal synapsids, Guadalupian therapsids, and Lopingian therapsids

with 999 iterations for significance testing. Guadalupian basal synapsids were grouped separately from therapsids of the same period because this test was intended to evaluate the effects of the adaptive radiation of therapsids after Olson's extinction on their evolutionary rates (see above).

The evolutionary dynamics of skull shape, function, and size were characterised by fitting evolutionary models to these data (PC scores for shape, functional character data for function, log-transformed centroid size for skull size). As above, Brownian motion, early burst, kappa, and delta models were tested in R using the function *transformPhylo.ML* in the package motmot[66]. Model fit was assessed using both log-likelihood (lnL) and Akaike Information Criterion (AIC) values.

## Reporting summary

Further information on research design is available in the Nature Portfolio Reporting Summary linked to this article.

## Data availability

Occurrence and phylogenetic data from Singh et al.[5], which formed the basis for the corresponding datasets compiled here are available to download from Dryad[71]: https://doi.org/10.5061/dryad.jq2bvq8h0. Specimens examined during the construction of adjacency matrices for AnNA are accessioned within the collections of the University of Cambridge Museum of Zoology (*Dimetrodon* UMZC T.320) and Natural History Museum London (*Moschorhinus* NHMUK R5698 and *Procynosuchus* NHMUK R37054). All raw data required to reproduce analyses reported here, including a list of specimens included within morphometric analyses, are available to download from Figshare[72]: https://doi.org/10.6084/m9.figshare.31150477. Phylogenetic data, including raw data used to generate the time tree used here as well as the completed tree, are also available to download from Figshare[73]: https://doi.org/10.6084/m9.figshare.31150492.

## Code availability

All R code required to reproduce analyses reported here is available to download from Figshare[74]: https://doi.org/10.6084/m9.figshare.31150501.

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

## Acknowledgements

Special thanks to Mathew Lowe (University of Cambridge Museum of Zoology) and Michael Day (Natural History Museum London) for facilitating access to specimens used in this study. We also thank Christian Kammerer, Yuya Asakura, an anonymous reviewer, and the editor for comments that greatly improved the manuscript. This work was funded in part by E.A.W.'s MSc bench fees at the University of Bristol. Funded by NERC grant NE/X013111/1 (M.J.B.).

## Author contributions

E.A.W. collected and analysed the data and wrote the manuscript. E.A.W. and S.A.S. designed the study. S.A.S. and M.J.B. provided guidance and comments on the manuscript. All authors contributed to editing the manuscript and approved the final draft.

## Competing interests

The authors declare no competing interests.
