## [Transparent Peer Review file · Communications Biology]

Trophic ecology outweighed intrinsic constraints in shaping skull evolution of carnivorous Permian synapsids

Corresponding Author: Mr Elías Warshaw

Version 0:

Reviewer comments:

Reviewer #1

(Remarks to the Author)

Recently, Singh et al. (2024) used the morphometric and phylogenetic comparative methods to analyze the mandibular data, and charted carnivorous synapsid trophic morphology from the latest Carboniferous to the earliest Triassic. Following similar methods, this paper analyzed the skull shape, function and size of carnivorous synapsids. They show their shapes reflecting trophic function and similarities between distant relatives arising by convergence through natural selection. I generally agree with their claims, but think some places need consider more.

As the previous works, they also show high rates of shape evolution at the base of the Therapsida. However, it could be that the current fossil records do not show the real pattern. Recently, an oldest gorgonopsian was reported, and it indicates the hidden early history of therapsids (Matamales-Andreu et al., 2024). So the rate of shape evolution at the base of therapsids should be smaller than showed in Figure 4. The rate need to be reconsidered.

Matamales-Andreu, R., Kammerer, C. F., Angielczyk, K. D., Simões, T. R., Mujal, E., Galobart, À., & Fortuny, J. (2024). Early–middle Permian Mediterranean gorgonopsian suggests an equatorial origin of therapsids. *Nature Communications*, 15(1), 10346. <https://doi.org/10.1038/s41467-024-54425-5>

Also, here are some suggestion.

L266 synapsids or therapsids

L336 change Asia to Europe Also, Inostrancevia and rubidgeines (viz. Dinogorgon and Rubidgea) co-occurred in the Ruhuhu Basin.

Brant, A. J., & Sidor, C. A. (2023). Earliest evidence of Inostrancevia in the southern hemisphere: new data from the Usili Formation of Tanzania. *Journal of Vertebrate Paleontology*, 43(4), e2313622. <https://doi.org/10.1080/02724634.2024.2313622>

Reviewer #2

(Remarks to the Author)

I do not work with anatomical network analysis, and have some questions about the methodology. I do work with geometric morphometrics, in which for bilaterally symmetrical structures (such as the cranium), because left and right landmarks on opposing bones are correlated, we mirror one side of the skull and take the average positions of the involved paired landmarks. So I am a bit confused by some of the aspects of using both left and right sides for the network analysis. Why does the braincase (a median structure) cluster specifically with the left temporal region in your example for Dimetrodon in Figure 1? There is no indication of significant asymmetry (at least of biological origins, though taphonomic crushing is a problem) in these taxa, so shouldn't the left and right temporal regions be equally linked to the braincase? Why does the left jugal resolve as part of the snout/palate complex while the right jugal is part of the temporal region? This bone's connections are identical on the two sides of the skull, so this seems weird (it also conflicts with the colored skull image on the figure, in which the left jugal of Dimetrodon is shown as part of the temporal module). I read the referred Strong and Werneburg papers and could not find where this issue is addressed.

“Prevomer” should not be treated as a separate element in synapsids; it is homologous with the vomer. The majority of literature on “pelycosaurs” since ~1950 refers to these structures as the vomers, as does all therapsid literature since the 1900s. The vomers are paired in all carnivorous “pelycosaurs”, so it doesn’t make sense that you treat it as a single median element in your analysis. Dinocephalians like Titanophoneus have unfused left and right vomers, which should be treated separately based on the rest of your network protocol (therocephalians partially fuse the vomers, so that is a bit of a gray area).

What is the “otoccipital” in your network? Is this intended to represent the midline fusion of the prootic and opisthotic, as in *Inostrancevia*? In synapsids, the fusion of those bones is traditionally called the “periotic”, to distinguish it from the condition in, e.g., archosaurs where there are distinct prootic and otoccipital (formed by fusion of the opisthotic and exoccipital) bones. I am confused by the listed presence of this bone (“otoccipital”) in *Dimetrodon*, in which the prootics and opisthotics are all unfused, so following the rest of your protocol there should be left and right representatives of both bones.

Among your study taxa, the pterygoids only fuse to form a median element in gorgonopsians. In *Titanophoneus* and *Biarmosuchus*, they should be treated as separate left and right elements.

No splenials are listed for *Procynosuchus*, but these elements are still present in that taxon (and continue to be separate elements deeper into cynodont evolution).

Synapsid workers have generally taken to calling the dorsalmost median bone on the occiput the “postparietal” instead of the “interparietal”, based on developmental evidence that the interparietal of mammals arose by fusion of the postparietal and tabulars (Koyabu et al. 2012)

Overall I found this an interesting manuscript and a welcome addition to the body of work on synapsid functional diversity, but based on the problematic lack of consistency in treating the various paired cranial elements between taxa, would request the authors re-run the analyses with corrected data.

Sincerely,
Christian Kammerer

Some line edits:

Line 30: “The Permian (299–252 million years ago (Ma)) began in crisis, with increasing global aridity driving the collapse of the vast coal forests that had characterized the preceding Carboniferous (359–299 Ma), devastating terrestrial vertebrate assemblages which had theretofore maintained a strong reliance on water.”

This is not really an accurate representation of the timing of ecological change. The Carboniferous Rainforest Collapse was a pre-Permian phenomenon occurring between 307 and 303.7 million years ago in the Moscovian and Kasimovian (Pennsylvanian, late Carboniferous) (Wang et al., 2025). The broader trends under discussion here are accurate, but it is misleading to say that the Permian “began in crisis”, as terrestrial vertebrate faunas are fairly stable across the Gzhelian-Asselian boundary, with many genera (e.g., *Dimetrodon*, *Edaphosaurus*, *Ophiacodon*, *Sphenacodon*) known from both the latest Carboniferous and early Permian. This supports the idea that the “crisis” happened before the Permian started and terrestrial vertebrates had already adapted to the changes (by contrast, there is substantial turnover in marine systems at this time).

Line 114: “evolution is subject to both a variety...” is awkwardly worded, I would change this to simply “evolution is subject to a variety of structural as well as functional constraints”

Figure 3: The limited sampling in the Gzhelian creates an artefactually thin shape space. If you are basing this on actual specimens, I understand that the better-preserved and more complete skulls of *Sphenacodon* and *Ophiacodon* are from the Permian, but both of these genera are known from this age. The total absence of sphenacodonts in your first time slice is misleading, as you have taxa well-known from complete skulls (like *Haptodus*) earlier in the Kasimovian.

Line 334: “the gorgonopsian *Inostrancevia*, for example, evolved in Asia”—no fossils of *Inostrancevia* are known from Asia; the non-African specimens are from European Russia. Admittedly this is a modern sociopolitical and not a geological distinction, but the larger fact is that we don’t really know where *Inostrancevia* evolved because the fossil record is missing 99% of the Earth’s surface where terrestrial vertebrates were living in the Permian. It would be safer to say, “*Inostrancevia* and its closest relatives first appear in the fossil record in Laurasia, whereas rubidgeines are known only from southern Gondwana.”

References:

Koyabu, D. et al. 2012. Paleontological and developmental evidence resolve the homology and dual embryonic origin of a mammalian skull bone, the interparietal. *PNAS* 109: 14075-14080.

Wang, Y. et al. 2025. The Middle-Late Pennsylvanian event: Timing and mechanisms. *Palaeogeography, Palaeoclimatology, Palaeoecology* 667: 112893.

Reviewer #3

(Remarks to the Author)

It was a great honor to review this manuscript. The authors present compelling data concerning the various factors and constraints driving cranial evolution in Permian synapsids. Overall, the manuscript is well written, and the results are clearly presented. However, I have some concerns regarding the data collection. My main suggestions for the authors are as follows:

- What is the rationale for using only lateral views in the shape data for the 2D geometric morphometric analysis? Would it not be necessary to analyze ventral and dorsal views of the skulls, or even 3D morphology?
- P 16 L 567. The authors quantify bone connections in AnNA. Are these sources of the information based solely on images from the literature? If so, there may be limitations to the data collection. For instance, specimens where contacts between bones are not externally visible but are present internally may lead to inaccurate results if only images are used. To improve accuracy, it may be necessary to incorporate information from textual descriptions, use CT data, or conduct direct observations where possible.
- P 3 L 143. The authors mention that derived taxa tend to have lower N, K, and P and higher D and L values. It would be helpful to briefly explain what higher or lower values of these parameters indicate. Since, as seen later in the manuscript (page 13, line 408), the authors treat connection density as a proxy for integration, providing this context earlier in the manuscript would make it easier for readers to follow.
- P 6. As some of readers and I who are not a specialist in mammals, I found it difficult to identify which taxa are represented by the abbreviations of the symbols in Fig. 2 and 3, as well as which taxa are basal or derived. A more detailed explanation in the figure captions or in the Materials and Methods section would be appreciated. In addition, the significance of the results shown in the upper right portion of Figure 2 is unclear. It would be helpful for the authors to explain what is being conveyed. The meaning of the lighter and darker colors should also be clarified in the figure caption.
- It may be helpful for the authors to consider incorporating relevant previous studies on developmental modularity in the mammal skulls (e.g., Koyabu et al., 2014, Nature Communications), which seem not to have been cited in the current version of the manuscript.
- P 17 L 598. While the manuscript describes what each network parameter represents, it would be appropriate to also cite Esteve-Altava et al. (2013, Evolutionary Biology) in relation to network parameters.

It was a rewarding experience to contribute to this manuscript as a reviewer. If possible, I would be pleased to review a revised version in the future.

Yuya ASAKURA, The University of Tokyo.

Version 1:

Reviewer comments:

Reviewer #1

(Remarks to the Author)

I have no further comments on this version, but delete the empty table at L433

Reviewer #2

(Remarks to the Author)

Thank you for your detailed attention to my previous concerns. My original issues have been addressed and I believe the manuscript can now proceed to publication.

Reviewer #3

(Remarks to the Author)

I have reviewed the revised manuscript. The authors have effectively addressed the raised points, resulting in significant improvements. I have no further comments or requested revisions.

Response to reviewers

We thank the reviewers for their thoughtful and constructive comments. Our responses are provided below in red.

Reviewer #1 (Remarks to the Author):

Recently, Singh et al. (2024) used the morphometric and phylogenetic comparative methods to analyze the mandibular data, and charted carnivorous synapsid trophic morphology from the latest Carboniferous to the earliest Triassic. Following similar methods, this paper analyzed the skull shape, function and size of carnivorous synapsids. They show their shapes reflecting trophic function and similarities between distant relatives arising by convergence through natural selection. I generally agree with their claims, but think some places need consider more. Many thanks for the positive remarks, and for the very helpful suggestions, all of which we follow.

As the previous works, they also show high rates of shape evolution at the base of the Therapsida. However, it could be that the current fossil records do not show the real pattern. Recently, an oldest gorgonopsian was reported, and it indicates the hidden early history of therapsids (Matamales-Andreu et al., 2024). So the rate of shape evolution at the base of therapsids should be smaller than showed in Figure 4. The rate need to be reconsidered.

Matamales-Andreu, R., Kammerer, C. F., Angielczyk, K. D., Simões, T. R., Mujal, E., Galobart, À., & Fortuny, J. (2024). Early–middle Permian Mediterranean gorgonopsian suggests an equatorial origin of therapsids. *Nature Communications*, 15(1), 10346. <https://doi.org/10.1038/s41467-024-54425-5>

In line with this comment, we created a new time tree with node ages constrained to match those provided in the cited study and re-ran all relevant analyses (see updated Materials & Methods).

Also, here are some suggestion.

L266 synapsids or therapsids

This section has been rewritten to accommodate updated results, so the suggested edit is no longer required.

L336 change Asia to Europe Also, *Inostrancevia* and *rubidgeines* (viz. *Dinogorgon* and *Rubidgea*) co-occurred in the Ruhuhu Basin.

Brant, A. J., & Sidor, C. A. (2023). Earliest evidence of *Inostrancevia* in the southern hemisphere: new data from the Usili Formation of Tanzania. *Journal of Vertebrate Paleontology*, 43(4), e2313622. <https://doi.org/10.1080/02724634.2024.2313622>

Asia has been changed to Laurasia (see reviewer 2 comments). We have also clarified the eventual coexistence of *Inostrancevia* and *rubidgeines*.

Reviewer #2 (Remarks to the Author):

I do not work with anatomical network analysis, and have some questions about the methodology. I do work with geometric morphometrics, in which for bilaterally symmetrical structures (such as the cranium), because left and right landmarks on opposing bones are correlated, we mirror one side of the skull and take the average positions of the involved paired landmarks. So I am a bit confused by some of the aspects of using both left and right sides for the network analysis. Why does the braincase (a median structure) cluster specifically with the left temporal region in your example for *Dimetrodon* in Figure 1? There is no indication of significant asymmetry (at least of biological origins, though taphonomic crushing is a problem) in these taxa, so shouldn't the left and right temporal regions be equally linked to the braincase? Why does the left jugal resolve as part of the snout/palate complex while the right jugal is part of the temporal region? This bone's connections are identical on the two sides of the skull, so this seems weird (it also conflicts with the colored skull image on the figure, in which the left jugal of *Dimetrodon* is shown as part of the temporal module). I read the referred Strong and Werneburg papers and could not find where this issue is addressed.

This aspect of the results has been discussed in the revised manuscript, with citations to additional studies recovering similar results also added. The figure has also been corrected to show the modular organisation of the left side of the skull in *Dimetrodon*.

"Prevomer" should not be treated as a separate element in synapsids; it is homologous with the vomer. The majority of literature on "pelycosaurs" since ~1950 refers to these structures as the vomers, as does all therapsid literature since the 1900s. The vomers are paired in all carnivorous "pelycosaurs", so it doesn't make sense that you treat it as a single median element in your analysis. Dinocephalians like *Titanophoneus* have unfused left and right vomers, which should be treated separately based on the rest of your network protocol (therocephalians partially fuse the vomers, so that is a bit of a gray area).

"Prevomer" has been corrected to "vomer" in all relevant adjacency matrices, and the vomers have been rescored as paired elements in sampled "pelycosaurs" as well as *Titanophoneus*. Following Durand (1991), we treat the partially fused vomers of the therocephalian *Moschorhinus* as a single element.

Durand, J. F. (1991). *A revised description of the skull of Moschorhinus (Therapsida, Therocephalia)*. South African Museum.

What is the "otoccipital" in your network? Is this intended to represent the midline fusion of the prootic and opisthotic, as in *Inostrancevia*? In synapsids, the fusion of those bones is traditionally called the "periotic", to distinguish it from the condition in, e.g., archosaurs where there are distinct prootic and otoccipital (formed by fusion of the opisthotic and exoccipital) bones. I am confused by the listed presence of this bone ("otoccipital") in *Dimetrodon*, in which the prootics and opisthotics are all unfused, so following the rest of your protocol there should be left and right representatives of both bones.

"Otocipital" has been corrected to "periotic" in all relevant adjacency matrices. For *Dimetrodon*, *Varanops*, and *Ophiacodon*, "otoccipital" was used to refer to the opisthotic-prootic-supraoccipital complex, described by Romer & Price (1940) as "a solid mass of bone (p.

64)” in which “no subdivisions can be seen (p. 64)”. We maintain the treatment of this structure as a single element given that Bazzana-Adams et al. (2023) could not discern sutures between its constituent bones in CT data of a specimen of *Dimetrodon*. However, “otoccipital” has been changed to “opisthotic-prootic-supraoccipital complex” in these taxa to distinguish it from the periotic of more derived synapsids.

Romer, A. S., & Price, L. W. (1940). *Review of the Pelycosauria* (Vol. 28). Geological Society of America.

Bazzana-Adams, K. D., Evans, D. C., & Reisz, R. R. (2023). Neurosensory anatomy and function in *Dimetrodon*, the first terrestrial apex predator. *Isience*, 26(4).

Among your study taxa, the pterygoids only fuse to form a median element in gorgonopsians. In *Titanophoneus* and *Biarmosuchus*, they should be treated as separate left and right elements. **The pterygoids have been rescored as paired elements in *Titanophoneus* and *Biarmosuchus*.**

No splenials are listed for *Procynosuchus*, but these elements are still present in that taxon (and continue to be separate elements deeper into cynodont evolution).

Splenials have been added to the adjacency matrix of *Procynosuchus*.

Synapsid workers have generally taken to calling the dorsalmost median bone on the occiput the “postparietal” instead of the “interparietal”, based on developmental evidence that the interparietal of mammals arose by fusion of the postparietal and tabulars (Koyabu et al. 2012) **“Interparietal” has been corrected to “postparietal” in all adjacency matrices.**

Overall I found this an interesting manuscript and a welcome addition to the body of work on synapsid functional diversity, but based on the problematic lack of consistency in treating the various paired cranial elements between taxa, would request the authors re-run the analyses with corrected data.

Many thanks for these positive remarks.

Sincerely,
Christian Kammerer

Some line edits:

Line 30: “The Permian (299–252 million years ago (Ma)) began in crisis, with increasing global aridity driving the collapse of the vast coal forests that had characterized the preceding Carboniferous (359–299 Ma), devastating terrestrial vertebrate assemblages which had theretofore maintained a strong reliance on water.”

This is not really an accurate representation of the timing of ecological change. The Carboniferous Rainforest Collapse was a pre-Permian phenomenon occurring between 307 and 303.7 million years ago in the Moscovian and Kasimovian (Pennsylvanian, late Carboniferous) (Wang et al., 2025). The broader trends under discussion here are accurate, but it is misleading to say that the Permian “began in crisis”, as terrestrial vertebrate faunas are fairly stable across the Gzhelian-Asselian boundary, with many genera (e.g., *Dimetrodon*, *Edaphosaurus*,

Ophiacodon, Sphenacodon) known from both the latest Carboniferous and early Permian. This supports the idea that the “crisis” happened before the Permian started and terrestrial vertebrates had already adapted to the changes (by contrast, there is substantial turnover in marine systems at this time).

The timing of discussed events has been corrected.

Line 114: “evolution is subject to both a variety...” is awkwardly worded, I would change this to simply “evolution is subject to a variety of structural as well as functional constraints”

The suggested change has been made.

Figure 3: The limited sampling in the Gzhelian creates an artefactually thin shape space. If you are basing this on actual specimens, I understand that the better-preserved and more complete skulls of Sphenacodon and Ophiacodon are from the Permian, but both of these genera are known from this age. The total absence of sphenacodonts in your first time slice is misleading, as you have taxa well-known from complete skulls (like Haptodus) earlier in the Kasimovian.

The figure caption has been updated to reflect this point, and to explain biases in the figure.

Line 334: “the gorgonopsian Inostrancevia, for example, evolved in Asia”—no fossils of Inostrancevia are known from Asia; the non-African specimens are from European Russia. Admittedly this is a modern sociopolitical and not a geological distinction, but the larger fact is that we don’t really know where Inostrancevia evolved because the fossil record is missing 99% of the Earth’s surface where terrestrial vertebrates were living in the Permian. It would be safer to say, “Inostrancevia and its closest relatives first appear in the fossil record in Laurasia, whereas rubidgeines are known only from southern Gondwana.”

The suggested change has been made.

References:

Koyabu, D. et al. 2012. Paleontological and developmental evidence resolve the homology and dual embryonic origin of a mammalian skull bone, the interparietal. PNAS 109: 14075-14080.

Wang, Y. et al. 2025. The Middle-Late Pennsylvanian event: Timing and mechanisms. Palaeogeography, Palaeoclimatology, Palaeoecology 667: 112893.

Reviewer #3 (Remarks to the Author):

It was a great honor to review this manuscript. The authors present compelling data concerning the various factors and constraints driving cranial evolution in Permian synapsids. Overall, the manuscript is well written, and the results are clearly presented. However, I have some concerns regarding the data collection.

Many thanks for the positive appreciation of our paper.

My main suggestions for the authors are as follows:

- What is the rationale for using only lateral views in the shape data for the 2D geometric

morphometric analysis? Would it not be necessary to analyze ventral and dorsal views of the skulls, or even 3D morphology?

We avoided using 3D data as this would have severely restricted the size of our dataset. This point has been clarified in the revised Methods section. Furthermore, using only lateral views for shape analyses allowed us to minimise overlap in the morphology recorded by 2D GM and analyses of functional characteristics. This point is highlighted in the Methods section under the subheading “*Functional morphometric analyses.*”

- P 16 L 567. The authors quantify bone connections in AnNA. Are these sources of the information based solely on images from the literature? If so, there may be limitations to the data collection. For instance, specimens where contacts between bones are not externally visible but are present internally may lead to inaccurate results if only images are used. To improve accuracy, it may be necessary to incorporate information from textual descriptions, use CT data, or conduct direct observations where possible.

In addition to images, textual descriptions and personal observations of specimens were utilised in data collection. This has been clarified in the revised manuscript, and sources of data as well as specimens observed in person have been cited.

- P 3 L 143. The authors mention that derived taxa tend to have lower N, K, and P and higher D and L values. It would be helpful to briefly explain what higher or lower values of these parameters indicate. Since, as seen later in the manuscript (page 13, line 408), the authors treat connection density as a proxy for integration, providing this context earlier in the manuscript would make it easier for readers to follow.

The suggested change has been made.

- P 6. As some of readers and I who are not a specialist in mammals, I found it difficult to identify which taxa are represented by the abbreviations of the symbols in Fig. 2 and 3, as well as which taxa are basal or derived. A more detailed explanation in the figure captions or in the Materials and Methods section would be appreciated. In addition, the significance of the results shown in the upper right portion of Figure 2 is unclear. It would be helpful for the authors to explain what is being conveyed. The meaning of the lighter and darker colors should also be clarified in the figure caption.

The suggested change has been made.

- It may be helpful for the authors to consider incorporating relevant previous studies on developmental modularity in the mammal skulls (e.g., Koyabu et al., 2014, Nature Communications), which seem not to have been cited in the current version of the manuscript.

The suggested change has been made.

- P 17 L 598. While the manuscript describes what each network parameter represents, it would be appropriate to also cite Esteve-Altava et al. (2013, Evolutionary Biology) in relation to network parameters.

The suggested change has been made.

It was a rewarding experience to contribute to this manuscript as a reviewer. If possible, I would be pleased to review a revised version in the future.

Yuya ASAKURA, The University of Tokyo.

We thank the reviewers for their timely and constructive reviews of our manuscript. Our responses are provided below in red.

Reviewer #1 (Remarks to the Author):

I have no further comments on this version, but delete the empty table at L433

Many thanks for your careful review. The empty table was only present in the tracked changes document and is not included in the revised manuscript.

Reviewer #2 (Remarks to the Author):

Thank you for your detailed attention to my previous concerns. My original issues have been addressed and I believe the manuscript can now proceed to publication.

Many thanks for your careful review.

Reviewer #3 (Remarks to the Author):

I have reviewed the revised manuscript. The authors have effectively addressed the raised points, resulting in significant improvements. I have no further comments or requested revisions.

Many thanks for your careful review.